# What Quality-of-Life Dimensions Are Most Important to Older Adults from Culturally and Linguistically Diverse Backgrounds Receiving Aged Care Services? An Exploratory Study

**DOI:** 10.3390/geriatrics7060144

**Published:** 2022-12-19

**Authors:** Claire Hutchinson, Jenny Cleland, Ruth Walker, Julie Ratcliffe

**Affiliations:** 1Health and Social Care Economics Group, College of Nursing and Health Sciences, Flinders University, Adelaide, SA 5001, Australia; 2Caring Futures Institute, Flinders University, Adelaide, SA 5001, Australia; 3Disability and Community Inclusion, College of Nursing and Health Science, Flinders University, Adelaide, SA 5001, Australia

**Keywords:** aged care services, culturally and linguistically diverse, long-term care, older adults, quality of life

## Abstract

There is little research on what aspects of quality of life (QoL) are most important to culturally and linguistically diverse (CALD) older adults. This study aimed to identify what QoL dimensions were most important to CALD older adults receiving aged care services, and therefore, how relevant a new six dimensions QoL instrument developed for use in aged care is to this population. A three-stage, mixed-methods study was undertaken. Stage 1: n = 3 focus groups with aged care providers. Stage 2: n = 30 semi-structured interviews with Italian-born older adults in ethno-specific residential aged care. Stage 3: survey of n = 63 older adults from mixed CALD backgrounds receiving community aged care services. Overall, older adults asserted the importance of the six dimensions of the new QoL instrument. The importance of ‘identity’ and ‘purpose and meaning’ were identified via the focus groups; however, the community-based CALD older adults identified these aspects of quality of life as more important than older Italians in residential care. Being in ethno-specific residential aged care where needs relating to language, food, and religion were met and they continued to live with others from their community may have meant that the meeting of cultural needs was more taken for granted.

## 1. Introduction

Due to immigration policies to increase the population since the end of the Second World War, Australia has one of the largest percentages of foreign-born inhabitants at 29.9% [1]. As a result, 37% of adults aged 65 years and over are from culturally and linguistically diverse (CALD) backgrounds [2], and this population is growing increasingly diverse [3]. The aged care needs of CALD older adults have been met through ethno-specific facilities; a preference of older adults and their families [4,5,6]. However, due to the increasing diversity of this population and the growth of multiculturalism policies, it is increasingly expected that such needs can be met through mainstream aged care services [7,8].

### Quality of Life in Aged Care

Quality of life is a key outcome in aged care, especially when the main purpose of care may no longer be longevity. In its final report, the Royal Commission recommended the routine collection of quality-of-life data in both residential and community care [9]. Yet there has been limited qualitative research in Australia and internationally on what aspects of quality of life are most important to older adults from CALD backgrounds. In terms of quantitatively measuring quality of life, the development of some quality-of-life instruments has been based on qualitative work with older adults, though this has typically been conducted with predominantly native English speakers [10,11,12,13]). Furthermore, the most common quality-of-life instrument applied internationally across health and social care is the EQ-5D [14] which was developed with younger populations and therefore may not be reflective of the quality-of-life dimensions viewed as most important by older adults [15].

In recent years, our team has developed a new quality-of-life instrument with older adults accessing aged care services in residential aged care and in the community—the Quality-of-Life Aged Care Consumers (QOL-ACC) [10,12,15,16,17]. Although CALD older adults were included in the various stages of development, due to budget constraints only older adults who were able to understand and communicate in the English language were included. The relevance of the six included dimensions (mobility, pain management, independence, social connections, emotional well-being and activities) therefore needs to be tested on more diverse populations.

Most studies of CALD older adults in developed English-speaking nations have focused on Asian populations. In a qualitative study with older Chinese adults in Australia, Tsang and colleagues [18] identified that health, independence, financial security, having a meaningful role, strong ethnic support from the community and family, and love and respect were important aspects of quality of life. Another study of older Chinese adults in New Zealand identified the importance of maintaining independence and having strong family support [19]. Several studies have highlighted the importance of social connections with their ethnic community as important to older CALD adults, such as studies with older Punjabi women and older Korean adults in Canada [20,21]. Jo and colleagues [21] identified that having a sense of belonging and feeling valued were important to older Korean adults. In a Canadian study of older south-east Asian women investigating key factors impacting on quality of life, dimensions were negatively expressed as loneliness/social isolation (with limited English a compounding factor), experiences of abuse (physical, mental, economic) exacerbated by dependency, and expectations around family roles (which included the exchange of domestic labour and childcare for being housed and cared for). Similar factors were also expressed by older Chinese adults living in Australia, especially women, as impacting on their quality of life [22]. 

Therefore, this study aimed to identify (1) what QoL dimensions are most important for CALD older adults to feel like they are having a good QoL as they age, and (2) how relevant the six dimensions of the QOL-ACC are to this population. These research aims are addressed using a mixed methods approach with focus groups with aged care providers, and interviews and a survey with older CALD adults, both in residential aged care and living in the community. 

## 2. Materials and Methods

This study received ethics approval from the Flinders University Human Research Ethics Committee (reference: 2369). All participants provided informed consent.

### 2.1. Research Design and Impact of COVID-19

This mixed methods study was originally designed to include focus groups with aged care providers, and interviews with older adults from up to five different CALD communities living in residential aged care (n = 30). As the COVID-19 pandemic started shortly after the funding was received, it was extremely challenging to engage with providers and to obtain support for recruitment. As restrictions eased in South Australia, we received support from two ethno-specific providers to recruit older Italians, resulting in all the intended interviews being conducted in the Italian community. In order to diversify the perspectives, we produced a short survey on the importance of a number of quality-of-life dimensions identified in the focus groups and interviews and engaged with community-based organisations serving CALD older adults receiving services in the community to distribute the survey. These organisations were in two Australian states (South Australia and Queensland). 

### 2.2. Stage 1: Focus Groups with Providers

#### 2.2.1. Participants

Aged care providers in the authors’ networks were approached to participate in focus groups. A consultant with considerable networks in the aged care sector in South Australia was engaged to provide introductions to further organisations outside these personal networks. Focus group participants needed to be working in roles that put them in contact with CALD consumers or involved in the development of policy and practice frameworks for serving the needs of this consumer group. Once initial agreement from the organisation’s contact person had been given, the contact person at each organisation was sent participant information sheets and consent forms to distribute to potential participants. Once consent forms had been returned to the first author, focus groups were organised for a mutually convenience date and location. In total, n = 9 participants were recruited for three focus groups. 

#### 2.2.2. Materials

A focus group guide was developed by the first author and the aged care consultant to guide the discussion and circulated to the wider research team for comment. The guide consisted of the following sections. Firstly, participants were asked to provide some basic socio-demographic questions such as gender, age, country of birth, language spoken at home, years working in the aged care sector, highest level of education and their current role. 

Participants were then asked some open questions with prompts about what they perceived as most important for CALD older Australians to experience a good quality of life as they aged. Example questions include the following: What do you think is most important for older CALD people to feel like they are having a good life as they age? In what ways do you think that CALD older people’s quality of life can be enhanced by culturally appropriate services? 

Following that discussion, participants were presented with the quality-of-life conceptual model used to developed the QOL-ACC [10,12]. Participants were prompted to consider the relevance and relative importance of these quality-of-life dimensions to CALD populations in their facilities, and asked to reflect if there was anything else missing from the model that they perceived as important for CALD older adults.

#### 2.2.3. Analysis

The focus groups were audio recorded with the permission of participants and transcribed in full by a professional transcription service under contract to Flinders University. After reading the transcripts, the first author mapped the topics under discussion and these were subsequently reviewed by the authorship team. The topics that related to quality-of-life dimensions already captured under the existing QOL-ACC model were identified. The remaining content was reviewed and consolidated into additional themes or dimensions. These themes informed the development of quality-of-life cards for stage 2. 

### 2.3. Stage 2: Interviews with Italian-Born Older Adults

#### 2.3.1. Participants

Participants were recruited from four residential aged care homes operated by two residential aged care organisations. Eligibility to participate in the research was based on the following criteria: aged 65 years and over, living in residential aged care, born outside of Australia, from a CALD background and ability to provide informed consent. Convenience sampling was adopted with all older adults who met the criteria and wished to participate being interviewed. All participants were born in Italy. As older adults of Italian heritage typically have relatively low rates of proficiency in English [5], a professional Italian interpreter attended all interviews. 

The aged care organisations initially approached potential participants who met the inclusion criteria and distributed participant information sheets to interested residents. Residents interested in participating agreed to have their names added to a list sent to the research team. Formal consent was sought on the day the interviewer attended each facility. Participant information sheets and consent forms were presented to participants in Italian following translation of the documents from English by a professional translation company. Participants were able to attend the interview with a family member or member of staff if they wished. In total, n = 30 older adults were recruited. 

#### 2.3.2. Materials

A semi-structured interview schedule was developed to guide discussions and consisted of three sections. In the first section, participants were asked for some basic socio-demographic details (date of birth, gender, year of migration to Australia, time residing at facility). Participants were also asked if their culture, national heritage or faith had been a key factor in choosing their aged care facility, and finally they were asked to rate their health on a five-point scale (excellent, very good, good, fair, poor) [23]. In the second section, the interviewer asked open-ended questions about what was important for the participant to feel like they were having a good quality of life as they aged, with follow-up questions and prompts based on participants responses. These questions were adapted from the interview guide using in the first stage of development of the QOL-ACC [10]. Example questions: Tell me about what quality of life means to you? What do you think is most important for you to feel like they are having a good life? What things make you happy? What do you most value in life? 

The third part of the interview consisted of the presentation of a series of cards reflecting different dimensions of quality of life. The cards were based upon the six quality-of-life dimensions of the QOL-ACC [10,11,12] with additional dimensions identified from the analysis of focus group data (Stage 1). Each card displayed a single quality-of-life dimension with a brief description of that dimension. All cards were in Italian following professional translation. Participants were asked to select three cards or more that were most important to their quality of life. The cards were used to understand the quality-of-life characteristics the participants deemed most important, and to initiate further discussion about their quality of life. A similar approach was adopted for the first stage of development of the QOL-ACC and demonstrated that the use of cards extended the discussion on what was important for them to feel like they were having a good quality of life. 

#### 2.3.3. Analysis

All interviews were recorded with the permission of participants and transcribed in full by a professional transcription service under contract to Flinders University. Transcripts were imported into NVivo software Version 12 [24]. The analysis was guided by the purpose of the research which was to determine what quality-of-life characteristics were important to CALD older adults. A priori codes were established based on the quality-of-life cards generated from stage 1, with new codes created as required. The analysis was performed by the second author (JC) with codes reviewed by the first author (CH). The socio-demographic data were entered onto SPSS, Version 25.0 to generate frequencies and percentages. For the card task, percentages were created for each card (QoL dimension) selected over the total number of cards selected, as it was not anticipated that all participants would be able to select and rank three cards based on previous experiences of conducting such exercises with older adults. 

### 2.4. Stage 3: Survey of Older Adults from Mixed CALD Backgrounds

#### 2.4.1. Participants

Participants were recruited from two community organisations serving the needs of older CALD adults receiving aged care services in the community. To be eligible, participants had to be born outside Australia, from a CALD community, over 65 years and receiving aged care services in the community. Convenience sampling was adopted with all older adults who met the criteria and wished to participate being included. Given the diverse cultural backgrounds and languages spoken, staff from the two organisations agreed to support survey completion if required, as it was not possible to translate the survey into multiple languages due to budgetary constraints. The surveys were distributed over multiple days with some events served by bi-lingual staff members and some older adults able to self-complete due to being proficient in English. In total, n = 63 participants were recruited. 

#### 2.4.2. Materials

The survey consisted of the quality-of-life dimensions and descriptions used in the card task in stage 2. For each dimension, participants were asked to circle a number ranging from ‘1’ (not at all important) to ‘10’ (very important) to reflect how importance each quality-of-life dimension was to their own quality of life. The survey also included some socio-demographic questions and some questions about the participants CALD background. 

#### 2.4.3. Analysis

Data from the survey were entered onto SPSS, version 25.0 and descriptive statistics conducted on socio-demographic data and the importance scores (1 to 10) for each quality-of-life dimension. 

## 3. Results

### 3.1. Stage 1: Focus Groups with Providers

Three organisations agreed to participate in focus groups. Focus group 1 was with an ethno-specific aged care provider (‘ethno-specific’, n = 2), focus group 2 was with a provider that had an ethno-specific unit at one of its facilities but otherwise offered mainstream aged care services (‘hybrid’, n = 3), and focus group 3 was with a provider that only offered mainstream services (‘mainstream’, n = 4). The focus groups were conducted by a facilitator and the first author. 

Socio-demographic details of the focus group participants are shown in Table 1. Participants were generally highly experienced in aged care, well-educated and came from diverse cultural backgrounds. Two were in senior management roles, and two in operational roles (one managing an ethno-specific unit and one a site manager). The four participants in project officer or co-ordinator roles were working in areas related to diversity as well as mental health and well-being.

In reflecting on what aspects of quality of life they thought were most important to older CALD adults, participants focused exclusively on psychosocial aspects of quality of life. Once the QOL-ACC model was presented, participants agreed that health-related aspects of quality of life—mobility and pain management—were also important; however, these QOL dimensions were not elicited in the open questions section of the focus group.

Of the other QOL-ACC domains, social connections was the dimension that was most discussed. Participants highlighted the centrality of the family unit to CALD older adults, irrespective of cultural heritage. As one participant from an ethno-specific aged care facility, which serves a European community, states: “*it’s all about the family*” (Participant 1, Ethno-specific). The unit manager of the ethno-specific unit in a mainstream service serving an Asian community echoed this with a similar comment: *I can say all their family, children, everything connect to them, that’s very, very important.* (Participant 3, Hybrid) 

Being with others who speak their language was viewed as very important. Many residents in the ethno-specific service knew each other before migration as well as before moving into residential aged care. For residents of the mainstream services, the importance of social clubs in the community to facilitate these connections, and continued use of native language, were identified. 

Participants also affirmed the importance of independence, emotional well-being and activities to CALD older adults. 

For the unit serving an Asian population, the importance of exercise-based activities was particularly highlighted. 


*The Asian community they keep themselves very independent and fit, they do a lot of exercise culturally. So, we see them really enjoying when we do an exercise class…. So certainly, that’s a big thing that they love.*
(Participant 2, hybrid)

As with the Anglo populations, activities that reflected their previous lives and affirmed talents were viewed as meaningful activities for CALD older adults to engage in.


*We had a resident who was Vietnamese who was in our memory support unit with quite severe dementia, get her on the piano and she was just amazing. So, there’s someone that could not be able to communicate or cognitively quite declined, but she was amazing on the piano.*
(Participant 2, hybrid)

QOL dimensions that were identified over and above those included in the QOL-ACC were ‘identity’, and ‘purpose and meaning’. Participants viewed particular activities as supporting the celebration and affirmation of identity. In the ethno-specific facilities, the celebration of religious festivals, particularly saints from their own countries, were seen as affirming both cultural and religious identity. In the ethno-specific unit, the celebration of Vietnamese culture was actively participated in and enjoyed by the residents of that culture.

It was noted that the décor of the ethno-specific unit, in bedrooms and common spaces, was reflective of the culture of the residents in the unit and very distinct from the rest of the facility. 

Purpose and meaning were viewed as very important for older adults from CALD backgrounds: “*Well, I mean if you don’t have something meaningful what’s the point of getting up or existing?*” (Participant 1, Hybrid). Some participants acknowledged that religion provided purpose and meaning to some residents, whilst others saw purpose as contributing to society.


*That real sense of purpose and having a role or being felt to be contributing and being important to people I think is a really important thing.*
(Participant 3, Mainstream)

Cards relating to these two additional dimensions were added to the cards for the six dimensions of the QOL-ACC and presented to participants in Stage 2 (Table 2). 

### 3.2. Stage 2: Interviews with Italian-Born Older Adults

Interviews were conducted with 30 residents in four ethno-specific aged care facilities that serve the Italian community in South Australia. Participants’ details are shown in Table 3 below. The mean age of participants was 89.3 years (range 76 to 100 years). Only one participant spoke English fluently; all other interviews were conducted with the support of an Italian interpreter. Overall, 61.5% reported that their cultural background has been a factor in choosing their facility. 

Amongst participants, we found strong support for the importance of the six dimensions that make up the QOL-ACC. Participants also discussed the importance of the two additional dimensions identified in the focus groups: identity, and purpose and meaning. No additional themes were identified. 

Regarding independence, like the predominantly Anglo sample interviewed previously, participants stressed the importance of being able to do as much as possible themselves, especially around self-care. 


*So, it’s important. The less you’re dependent on people, the better you are. The more you can do, the better off you are.*
(Participant 24)

Participants valued being mobile and being supported by staff to stay mobile for as long as possible. 


*I’m lucky that I actually can walk. I don’t need a frame, but I have to have someone sort of following me.*
(Participant 7)

Participants reported health conditions that resulted in pain and highlighted the importance of making sure this was managed appropriately whether by medication or non- pharmaceutical options. 

Participants valued being “*emotionally well*” (Participant 25) and “*being happy with yourself*” (Participant 20). 

As with the Anglo sample, participants did not use terms such as mental health, depression or anxiety. For many of participants, emotional well-being was associated with having strong family connections. 


*It’s not always easy to be emotionally well, especially if you’re away from your family … spending a lot of time on your own is not really good for you.*
(Participant 25)

The importance of social connections was talked about extensively in relation to family and friends, as well as relationships with volunteers, staff and the broader community. 


*So, social contact is important, because people bring us information from outside… we live in a bit of a bubble, so having social contact is important to share information.*
(Participant 20)

As with the Anglo sample, participants valued engagement in activities, whether alone—such as crosswords and reading—or as part of a group—such as bingo. The quote below highlights the impact of COVID-19 on group activities within the facility.


*Prior to coronavirus, there was a lot of involvement from the community outside. So, we had a lot of school groups coming in, little entertainment groups, singing groups …There was a constant ebb and flow of people coming and going, entertaining us, doing activities with us…that’s very, very important for quality of life. Since coronavirus, that’s stopped.*
(Participant 20)

Participants affirmed the importance of their cultural identity to their quality of life. This was discussed as ‘knowing who you are’ and psychologically maintaining an identity as an Italian person. 


*Knowing who we are and our own personal—yeah, exactly. So I try and look after … who I am, my cultural being. And to do that is to help me not only physically but also mentally to know who I am.*
(Participant 24)

Purpose and meaning were not mentioned by many of the participants but, when it was, it was closely associated with cultural identity, being independent, and their role within their family. For a few participants, meaning was attached to their faith. 


*My family and my religion. The religion is the most important one for me, because I’m serving the grand god of the universe… All the rest come after that. My family comes after that.*
(Participant 8)

When presented with the cards and asked to identify the three quality-of-life dimensions that contributed the most to their quality of life, five participants reported that all the dimensions were equally important. Twenty-five selected at least one card, 23 selected two cards and eight selected three cards, making 63 cards in total. The most selected cards were social connections (n = 15, 24%), and independence (n = 12, 19%), followed by emotional well-being (n = 8, 13%), activities (n = 7, 11%), and pain (n = 7, 11%). Identity and purpose and meaning were only selected by two participants each (4%). Against expectation, mobility was only selected as important by two participants (4%). 

### 3.3. Stage 3: Survey of Older Adults from Mixed CALD Backgrounds

Sixty-three participants who were born in 23 different countries completed the survey. Participants ranged in age from 67 to 96 years (mean = 80.6 years) and 78% were female. Overall, when asked about the importance of providers being able to meet their cultural needs, 13% of respondents reported it was ‘important’ and 64.8% ‘very important’. Only 5.6% reported that it was not important. On average, participants had been accessing aged care services in the community for seven years. 

Participants were presented with the six QOL-ACC dimensions and the two additional dimensions identified in the focus groups with providers and validated as important to at least some of the interviewees in study 2. These were presented with definitions and a 10-item scale of ‘1’ (not important to their quality of life) to ‘10’ (very important to their quality of life) (Table 4). 

All dimensions were viewed as important for a good quality of life, with even the lowest mean score (pain management) receiving a mean score of 8.2 out of 10. The dimensions with the highest proportion of respondents scoring ‘10’ were independence (65.6%) and identity (62.1%). 

## 4. Discussion

This study explored what quality-of-life dimensions are most important to older adults from CALD backgrounds as they age, with a secondary aim of identifying if the six dimensions of the QOL-ACC were also important to older adults from CALD backgrounds. Across both the Italian residential care sample and the multicultural sample of older adults receiving aged care services in the community, the importance of the six QOL-ACC dimensions was supported as important. 

The two additional dimensions identified initially by aged care providers—‘identity’ and ‘purpose and meaning’—were also found to be important to older adults from diverse CALD backgrounds. However, those living in the community identified these aspects of quality of life as more important than the older Italians living in residential care. The older Italian adults predominantly asserted the importance of the six QOL-ACC dimensions over the two additional quality-of-life dimensions. For the older Italians, being in ethno-specific residential aged care which is set up to meet needs relating to language, food, and religion, and where they continued to live with many others from their wider cultural community may have meant that, to some extent, the meeting of cultural needs was more taken for granted and therefore less salient when discussing their quality of life. 

The older adults who completed the survey were in part supported in maintaining their cultural identity by attending the lunch clubs and other day services supported by the two community-based multicultural organisations where they completed the survey. Though it was noted that some nationalities—Filipino and Japanese—had specific events for their communities, given the diverse CALD backgrounds of older adults living in different Australian states [3,25], there may not be enough numbers to maintain specific events for different groups based on language, nationality or other characteristics. Older adults and their families in the community may have to make additional efforts to identify and engage with such activities to support ongoing use of language, access to culturally appropriate food or even just to maintain social connections with other older adults from the same community or with similar migration experiences. 

This research aligns with the small number of other studies conducted with migrant populations in Australia and internationally. Other studies have highlighted the importance of independence [18,19] and social connections [20,21] to older adults from CALD backgrounds. The quality-of-life dimension of having a meaningful role identified in a previous study [18] was similar to the dimension of ‘purpose and meaning’ identified in the current study. It is noted that in other international studies of non-migrant populations, the quality-of-life dimension of purpose and meaning has been identified as important [26,27]. In Drageset and colleagues’ Norwegian study, the key experiences that promoted feelings of meaning and purpose were identified as physical and mental well-being, belonging and recognition, personally treasured activities, and spiritual closeness and connectedness [26]. Notably, belonging related to being socially connected to family and friends as well as having good relationships with care staff. Treasured activities related to both gathering for enjoyable activities with others and engaging in solo activities. Similarly, a Swedish study identified that purpose could be enhanced by engaging in meaningful activities [27]. A Spanish study identified that meaningful activities form a part of the identity of older adults in residential care and promoted a sense of belonging [28]. 

These examples of how purpose and meaning can be created strongly align with the QOL-ACC dimensions of social connections and activities. These studies also highlighted the importance of having a spiritual life or belonging to a faith group [26,27]. Faith was identified as important to many of the CALD participants in the current study but was not identified as being significantly important to the predominantly Anglo samples of older adults using aged care services included in the various development stages of the QOL-ACC. Purpose in life can be measured using other existing instruments (e.g., [29]; however, aged care providers might be better placed to meet these particular needs of older CALD adults through discussion of what is personally meaningful to them. Furthermore, Identity is acknowledged to be a complex concept [30,31], and language, religion and other cultural factors may not align with nationality or country of origin. Therefore, what supports identity may be very personal to the individual. As with the six dimensions of the QOL-ACC, aged care services can make significant impacts on the quality of life of older CALD adults in relation to supporting identity, purpose and meaning. 

The interview data in particular highlighted the impact that the COVID-19 global pandemic has had on the quality of life of older adults in residential aged care which has vastly reduced outdoor and indoor activities for residents, as well as limited family and friends visiting facilities [32]. There was evidence from the interviews that the impacts on both activities and social connections of restrictions had negatively impacted on residents’ lives and, as experienced in Australia and internationally, increased the risk of social isolation [33,34]. It is notable from much of the discussion on quality of life from providers, residents and older adults in the community that there was a strong focus on psycho-social aspects of quality of life over health-related quality of life. This highlights that generic adult QoL instruments primarily focused on health-related QoL may not be the most appropriate for this population. Four of six QOL-ACCs dimensions focus on psycho-social QoL.

This research shows that older CALD adults have some differences in the quality-of-life dimensions they value relative to Anglo older populations in Australia, though a case for significant commonalities can be made based on this relatively modest sample. Routine measurement of quality of life would be a valuable quality indicator of aged care services which would support consumer choice. In addition, such data could be used to identify service needs at the facility level which would be valuable to aged care service providers as well as consumers and their families. 

### Strengths and Limitations

This study provides valuable data on the importance of quality-of-life dimensions to older CALD adults which is important if quality of life is to be routinely assessed in aged care as per the Royal Commission recommendation [9]. The study was able to capture the views of aged care providers as well as older CALD adults living in residential aged care and the community. Researchers suggest that a minimum of six participants is desirable to conduct a focus group [35]. However, during COVID-19 restrictions it was challenging to get any engagement in research from aged care providers. Consequently, the decision was taken to conduct the focus groups with whomever attended on the day rather than abandon aged care provider involvement in the research project. Whilst the survey enabled the inclusion of participants from a diverse range of cultural backgrounds, the quantitative approach of this stage of data collection meant that we were unable to explore additional quality-of-life dimensions that might also be important to this group, which may have been possible using interview methodology. Overall, it would have been beneficial to undertake interviews with a larger sample of older CALD adults in aged care from a range of different cultural backgrounds, as well as those living in mainstream, hybrid and other ethno-specific residential aged care facilities. However, small samples sizes for research with non-English speaking older adults is common given the challenge of recruiting and the costs of hiring interpreters. The studies conducted in other English-speaking nations with Asian older adults had interview sample sizes of between 5 and 30 [18,20], and a previous survey study had a sample size of 79 [21]. Additionally, as previously mentioned COVID-19 created significant barriers to gaining support for recruitment at a time when aged care providers had many additional demands placed on them. 

## 5. Conclusions

This study confirms the importance of the six dimensions of the QOL-ACC (mobility, pain management, independence, emotional well-being, social connections and activities) to this sample of older CALD adults living in residential aged care as well as in the community. The importance of older CALD adults maintaining their identity and having meaning and purpose in their lives is also important for them to feel like they are having a good quality of life as they age. To some extent, identity and purpose and meaning may be achieved through social connections to their community (including use of language), by engaging in meaningful activities, including important cultural and religious events, and by maintaining their emotional well-being. Given the complex nature of both these constructs, engagement with older adults and their families regarding what is personally meaningful to support identity and provide purpose and meaning is likely to enhance the quality of life of older CALD adults in aged care. Given the diversity of Australian’s CALD population, it is unlikely that ethno-specific aged care will be available for all CALD groups when they need aged care services. Therefore, mainstream aged care services will need to meet the needs of an increasingly diverse older population. It would be beneficial for future work to develop resources to support such discussions, and subsequent care plans, and for providers to share best practice to ensure that quality of life is maximised for older CALD Australians. 

## Figures and Tables

**Table 1 geriatrics-07-00144-t001:** Focus Group Participants socio-demographic details.

	Frequency
Gender	
Male	4
Female	5
Age	
30 to 39 years	2
40 to 49 years	2
50 to 59 years	2
60 years and over	2
Place of Birth	
Australia	3
Europe	4
Asia	1
Language spoken at home	
English	6
Other	3
Highest Qualification	
Vocational	1
Post-graduate	7
Years working in the aged care sector	
Less than 5 years	1
5 to 9 years	2
10 to 20 years	2
More than 20 years	3
Current role	
Senior Manager/Executive	3
Project Officer/Co-ordinator	4
Operational	2

NB: Some missing data on one participant.

**Table 2 geriatrics-07-00144-t002:** Quality-of-Life Dimension Cards.

QOL-ACC Dimensions	Additional Dimensions
Mobility Pain Management Independence Social Connections Emotional Well-being Activities	Identity Purpose and Meaning

**Table 3 geriatrics-07-00144-t003:** Socio-demographic data: interview participants.

	n (%)
Gender	
Male	8 (26.7)
Female	22 (73.3)
Age	
65 to 79 years	2 (6.9)
80 to 89 years	12 (41.4)
90 years and over	15 (51.7)
	**Mean (SD) Range**
Years Living in Australia	
Mean (SD)	67.2 (8.10)
Range	53–91
Years living in current location	
Mean (SD)	3.6 (3.52)
Range	0.2–15

Note. n = 30.

**Table 4 geriatrics-07-00144-t004:** Ranking of QoL Items.

	Range	Mean (SD)	% Very Important (10)
Identity	3–10	8.9 (1.76)	62.1%
Social Connections	5–10	8.8 (1.68)	54.8%
Purpose and meaning	4–10	8.8 (1.67)	55.2%
Independence	4–10	8.8 (1.98)	65.6%
Activities	4–10	8.7 (1.69)	47.5%
Emotional Wellbeing	3–10	8.7 (1.76)	45.9%
Mobility	3–10	8.3 (2.03)	47.5%
Pain Management	2–10	8.2 (2.04)	40.0%

## Data Availability

Data are contained within the article. Additional data are not available due to ethics requirements.

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
