# Peer review of "What Quality-of-Life Dimensions Are Most Important to Older Adults from Culturally and Linguistically Diverse Backgrounds Receiving Aged Care Services? An Exploratory Study"

_geriatrics, 2022, doi:10.3390/geriatrics7060144_

Round 1
Reviewer 1 Report (Previous Reviewer 4)
My congratulations to the authors, for taking on board the recommendations made in the 1st round of review and improving the article.
Best regards
Author Response
Thanks for your kind comments.
Reviewer 2 Report (New Reviewer)
The study is well conducted and captures the view of aged care providers. It also highlights the limitations of the study which would have been beneficial to undertake interviews with a larger sample, however it has to be acknowledged that Covid-19 had an impact.
Author Response
Thanks for your kind comments.
Reviewer 3 Report (New Reviewer)
This is a well written paper, targeting an important demographic. The three-part mixed-method approach makes it more complex to report and the authors have done a good job of presenting the three sections clearly. This mixed-methods design adds to the study. Adjustments made to the recruitment, as necessitated by the COVID-19 pandemic, appear to be appropriate to the aims of the research with efforts made to maintain participant diversity. The resultant limitation to the breadth of qualitative research participants is well discussed in 4.1.
To aid readability:
2.2.2 Materials
The example questions used in the focus groups could be put in a table or an appendix. Currently the paragraph is very wordy.
The 6 quality of life dimensions from the model are repeated from 1.1 paragraph 2. If these were in a box/table they could be easily referred back to by the reader.
2.3.2 Materials
As above, the description of the questions and example questions could be put in a table or an appendix making the paragraph more succinct.
3.3 Stage 3: Survey
Would you consider putting the survey instrument as an appendix?
Author Response
Thank you for your kind comments.
Thanks for your comments to help aid readability. We have:
- Removed repetition of the QOL-ACC dimensions in the text.
- We have added an extra table (Table 2) which lists the 8 QOL cards based on the six QOL-ACC dimensions plus the two additional dimensions.
- We have reduced the example questions in section 2.2.2 to two examples.
Given the survey was short and is described in full in section 2.4.2, we did not think adding an appendix to the paper would particularly enhance readers’ understanding.
Reviewer 4 Report (New Reviewer)
This study examined what quality of life dimensions are most important to older adults from culturally and linguistically diverse (CALD) backgrounds, with a secondary aim of identifying if the six dimensions of the QOL-ACC were also important to older adults from different backgrounds. The topic is current and interesting.
I have some notes. The number of participants was low in all groups. The description of the participant focus groups is complicated and difficult to understand. The conclusion is too general, not specifyed enough. QoL-ACC questions are also too general and it contains simple dimensions. On the other hand, the quotes of the participants improve and specify the conclusion.
Author Response
Thank you for your comments.
We acknowledge that the samples were small, and this has been identified in the limitations (section 4.1).
We have gone through the focus groups sections in the methods to streamline the language. In the materials section, we have broken up the key points into paragraphs. We hope this aids clarity.
We have extended the conclusions to tease out the implications and to suggest future work to support the quality of life of older CALD Australians.
The QOL-ACC was developed over a three year period and reflects what older adults identified as most important. The development of the dimensions, questions and psychometric testing are reported elsewhere and are not the subject of this paper.
We are glad that the quotes we used illustrate our points and we hope you enjoying reading the paper.
This manuscript is a resubmission of an earlier submission. The following is a list of the peer review reports and author responses from that submission.
Round 1
Reviewer 1 Report
Thank you for the opportunity to review this paper that adds to important understanding on QOL among CALD adults. The sample sizes were rather small for generalizations though confirmation of the 6 dimensions of the QOL-ACC is supported by the limited sample. The manuscript is well written.
Reviewer 2 Report
The study has three strong limitations:
1) the objectives presented in the manuscript do not seem to be the purpose of the study: instead, the authors seem to extend a previously developed and published instrument with two more dimensions (which have already been referenced in the literature). Furthermore, knowing the most important dimensions for older adults when presented with the instrument dimensions seems to correspond to knowing how relevant these dimensions are.
2) the methodology was not applied correctly (focus groups with 2 participants go against what has been recommended – see e.g., Sue Wilkinson (1998) Focus group methodology: a review, International Journal of Social Research Methodology, 1:3, 181-203, DOI: 10.1080/13645579.1998.10846874 ) and lacks information that demonstrates how several aspects were implemented:
- for example, the interview and survey guides;
- justification of the scales used: why is a scale from 0 to 10 the best for older adults?; and why were several scales used (in sample 2: important/not important in the cards; Likert-5 for health status)?;
- explanation of the sample in the focus group: why are executives appropriate since they are expected to have less contact with residents?
- data that is said to have been collected but then does not appear in the results (e.g., line 221 -223);
- explanation of how the data analysis was carried out (e.g., line 244 refers to the creation of new analysis codes when analyzing the content of the interviews - which ones?)
- It would be important to provide participants with clear definitions of what each dimension means, as different perspectives were assumed (line 432-434)
3) the size of the samples: the sample is so small that it is not possible to conclude what the authors refer to (especially given the huge percentage of migrants in Australia). This can be seen as an exploratory study, in which case it does will not contribute significantly to the literature.
Other comments:
- table 1. Frequency: 8 or 9 answers?
- line 270, "The survey also included some socio-demographic questions and some questions about the participants CALD background." - which ones?
Reviewer 3 Report
Thank you for your paper named "What quality of life dimensions are most important to older adults from culturally and linguistically diverse backgrounds receiving aged care services?". This is a great contribution to the current evidence on important dimensions of quality of life among CALD older adults.
Reviewer 4 Report
Dear authors,
Congratulations on the completion of this research work, for your time and dedication.
My comments are very positive about your research.
I congratulate you on the conceptualisation of the problem, the design and method, as well as the discussion of the debate and conclusion. Very elaborate.
I also congratulate you on the in-depth discussion and broad conceptualisation, as well as the detailed method followed.
I will now make some suggestions for improvement, with the aim of improving your citations, downloads, visits etc.
-I suggest that you incorporate as much as you can at the end of the discussion:
a) what are the theoretical implications of this work for the scientists who read this work, for the theoreticians in the field or colleagues.
b) what practical implications does this work have for older people?
c) future lines of research arising from this work that need to be pursued.
d) I suggest that you incorporate if you wish, a study of “In a study conducted on the quality of life of older people, the results showed that the components that most influence the quality of life are related to health, relationships, functional autonomy and staying active as the main factors influencing the quality of life of active older people compared to the economy, pension, housing or income” https://www2.uned.es/intervencion-inclusion/documentos/articulos/COMPONENTES%20DE%20INFLUENCIA%20MAS%20VALORADOS%20EN%20LA%20CALIDAD%20DE%20VIDA%20POR%20LAS%20PERSONAS%20MAYORES%20DE%2060%20ANOS%20FISICAMENTE%20ACTIVAS.pdf
e) Also if you wish to enrich your text: "Despite this, we must not forget that physical activity reinforces the mental, physical and social health dimensions and therefore the quality of life of older people".
https://doi.org/10.3390/ijerph17041299
I hope it will get better visibility this way! Congratulations!
My sincere congratulations on your work.